# Comparison of Bacterial Communities in Five Ectomycorrhizal Fungi Mycosphere Soil

**DOI:** 10.3390/microorganisms12071329

**Published:** 2024-06-29

**Authors:** Pi Chen, Zhen Li, Ning Cao, Rui-Xuan Wu, Zhao-Ren Kuang, Fei Yu

**Affiliations:** College of Forestry, Shanxi Agricultural University, Jinzhong 030801, China; chenpi25@outlook.com (P.C.); lizhen@sxau.edu.cn (Z.L.); s20222451@stu.sxau.edu.cn (N.C.); hachixuan1213@163.com (R.-X.W.); kzr1234562024@163.com (Z.-R.K.)

**Keywords:** ectomycorrhizal fungi, mycosphere soil, soil bacteria

## Abstract

Ectomycorrhizal fungi have huge potential value, both nutritionally and economically, but most of them cannot be cultivated artificially. To better understand the influence of abiotic and biotic factors upon the growth of ectomycorrhizal fungi, mycosphere soil and bulk soil of five ectomycorrhizal fungi (*Calvatia candida*, *Russula brevipes*, *Leucopaxillus laterarius*, *Leucopaxillus giganteus*, and *Lepista panaeola*) were used as research objects for this study. Illumina MiSeq sequencing technology was used to analyze the community structure of the mycosphere and bulk soil bacteria of the five ectomycorrhizal fungi, and a comprehensive analysis was conducted based on soil physicochemical properties. Our results show that the mycosphere soil bacteria of the five ectomycorrhizal fungi are slightly different. *Escherichia*, *Usitatibacter*, and *Bradyrhizobium* are potential mycorrhizal-helper bacteria of distinct ectomycorrhizal fungi. Soil water content, soil pH, and available potassium are the main factors shaping the soil bacterial community of the studied ectomycorrhizal fungi. Moreover, from the KEGG functional prediction and LEfSe analysis, there are significant functional differences not only between the mycosphere soil and bulk soil. ‘Biosynthesis of terpenoidsand steroids’, ‘alpha-Linolenic acid metabolism’, ‘Longevity regulating pathway-multiple species’, ‘D-Arginine and D-ornithine metabolism’, ‘Nitrotoluene degradation’ and other functions were significantly different in mycosphere soil. These findings have pivotal implications for the sustainable utilization of ectomycorrhizal fungi, the expansion of edible fungus cultivation in forest environments, and the enhancement of derived economic benefits.

## 1. Introduction

Not only are soil bacteria profoundly involved in biogeochemical cycling, they also play various key roles in many ecological functions [1]. Soil bacteria participate in the fixation, transformation, and release of nutrients through processes such as nitrogen cycling and phosphorus cycling [2]. Some bacteria engage in nitrification, converting ammonia nitrogen into nitrite and nitrate, which increases the nitrate nitrogen content in soil [3,4]. Soil bacteria also figure prominently in the phosphorus recycling process, promoting the effective utilization of phosphorus by degrading organic phosphorus and dissolving inorganic phosphorus [5]. Further, soil bacteria also play a pivotal role in organic matter’s decomposition and degradation [6,7,8], breaking it into inorganic salts and carbon dioxide that are released into soil [9].

In the belowground mutualism between mycorrhizas and plants, mycosphere soil bacteria play an important role in the strength of symbiosis between ectomycorrhizal fungi (ECM) and host plant species [10,11]. Garbaye [12] proposed the concept of mycorrhizal-helper bacteria (MHB). In forest ecosystems, there is an interdependent and coevolutionary relationship between these three entities: host plants, ectomycorrhizal fungi, and mycorrhizal auxiliary bacteria [13]. MHB can enhance the inoculation rate of mycorrhizal fungi [14], facilitate the germination of ectomycorrhizal fungal spores [15], and support their colonization of host plant roots to establish a strong mycorrhizal symbiotic structure [16]. Ectomycorrhizal fungi may host many MHB [17], such as *Pseudomonas* and *Bacillus* spp., whose presence can accelerate the growth of those ectomycorrhizal fungi and increase their overall production for harvesting [18]. There are complex relationships between ectomycorrhizal fungi and soil bacteria in the soil ecosystem. On the one hand, the presence of ectomycorrhizal fungi can change the soil bacterial community structure [19]. Bacteria can grow through the exudates of ectomycorrhizal fungi [20], but ectomycorrhizal fungi can also absorb bacteria as a source of nutrients [21]. On the other hand, soil bacteria can promote the growth and development of ectomycorrhizal fungi [22] or inhibit the growth of ectomycorrhizal fungi [23,24]. Most of the existing studies focus on soil bacteria of specific kinds of the ectomycorrhizal fungi mycosphere [13,25,26]. This study was extended to five ectomycorrhizal fungi to more comprehensively reveal the similarities and differences of soil bacteria of different ectomycorrhizal fungi mycospheres, and correlation analysis was carried out in combination with environmental factors. It provides a broader perspective for understanding the interaction between ectomycorrhizal fungi and soil bacteria.

The macrofungi *Calvatia candida*, *Russula brevipes*, *Leucopaxillus laterarius*, *Leucopaxillus giganteus*, and *Lepista panaeola* are all wild, edible, medicinal and ectomycorrhizal symbiotic fungi, harboring high economic and nutritional value. In this study, the mycosphere soil of ectomycorrhizal fungi collected from the Pangquangou National Nature Reserve was analyzed and compared to differences in soil physicochemical properties and soil bacterial community structure. The aim was to reveal the interactions between ectomycorrhizal fungi and soil bacteria and explore the effects of soil characteristics on the diversity and community structure of those bacteria. The findings provide a scientific foundation for the protection and reproduction of ectomycorrhizal fungi.

## 2. Materials and Methods

In August 2023, the mycosphere soil and bulk soil of five ectomycorrhizal fungi were collected from the mixed forest of *Picea asperata*—*Larix gmelinii* in the Pangquangou National Nature Reserve, located in Shanxi Province (China). The temperature range, soil type, longitude, latitude, and altitude of the site were recorded as well (Table 1). Here, the temperature is 22–25 °C, with brown earth as the soil type.

At the sampling point, the soil attached to the ectomycorrhizal fungi was collected and placed into a sterile polyethylene bag, this was termed ‘mycosphere soil’. At the same sampling point, the soil without any fungal fruiting body growth was collected 40 cm away from that ectomycorrhizal fungal specimen, and placed into a sterile polyethylene bag; this was called ‘bulk soil’. Three mycosphere soil samples paired with three bulk soil samples were collected at each sampling point for a total of 30 samples. All collected soil samples were divided into two parts and brought to the laboratory. One part was naturally air-dried and passed through a 2 mm sieve to measure various soil physicochemical properties. The other part was stored in a −70 °C freezer for subsequent molecular studies.

The air-dried samples were used to determine soil pH, utilizing a 2 mm mesh with a 1:2.5 (*w*/*v*) soil-to-water ratio suspension [27]. Soil organic carbon (SOC) was measured by the hot hydration method [28]. Available phosphorus (AP) was determined via double-acid extraction by the molybdenum-antimony resistance colorimetric method [29]. Soil available potassium (AK) was measured by flame photometry [30]. Available nitrogen (AN) was quantified by potassium persulfate oxidation [31]. Soil water content (SWC) was determined gravimetrically by drying at 105 °C for 6–8 h [32].

Total genomic DNA from each sample was extracted using the CTAB method. The DNA concentration and purity were monitored on 1% agarose gels to ensure an OD_260/280_ between 1.8 and 2.0 (for purity) and a DNA concentration >50 mg/L.

The bacterial regions (16S V3–V4) were amplified using a specific primer pair of 341F (5′-CCTAYGGGRBGCASCAG-3′) and 806R (5′-GGACTACNNGGGTATCTAAT-3′) [33]. Every PCR reaction was carried out with 15 µL of Phusion^®^ High-Fidelity PCR Master Mix (New England Biolabs, Beijing, China), 2 µM of forward and reverse primers, and about 10 ng of template DNA. The PCR reactions were as follows: 98 °C for 1 min, followed by 30 cycles at 98 °C for 10 s, 50 °C for 30 s, and 72 °C for 30 s, then ending with a final extension at 72 °C for 5 min (Bio-Rad T100 Thermocycler PCR system, Rocklin, CA, USA). 

Sequencing libraries were generated using NEB Next^®^ Ultra DNA Library Prep Kit (Illumina, San Diego, CA, USA) by following the manufacturer’s recommendations, with index codes added accordingly. Library quality was assessed on the Agilent 5400 system (Agilent Technologies Co., Ltd., Santa Clara, CA, USA) and each library was sequenced on the Illumina platform, generating 250 bp paired-end reads. The raw reads were deposited into the NCBI Sequence Read Archive (SRA) database (accession number: PRJNA1121956).

This analysis was conducted by following the ‘Atacama soil microbiome tutorial’ of Qiime2docs along with customized program scripts (https://docs.qiime2.org/2019.1/, accessed on 10 November 2023). Briefly, using the qiime tool import program, the raw data FASTQ files were imported into a format that could be operated by the QIIME2 pipeline system. Demultiplexed sequences from each sample were quality-filtered and trimmed, denoised, and merged, after which the chimeric sequences were identified and removed using the QIIME2 dada2 plugin to obtain the feature table of amplicon sequence variant (ASV) [34]. We then used the QIIME2 feature-classifier plugin to align the ASV sequences to a pre-trained SILVA v132 database at the 99% similarity level to generate the corresponding taxonomy table [35]. Diversity metrics were derived using the core-diversity plugin within QIIME2. Feature-level alpha diversity indexes, such as observed features, Chao1 richness estimator, and Shannon diversity index, were calculated to estimate the microbial diversity within an individual sample. In addition, the potential Kyoto Encyclopedia of Genes and Genomes (KEGG) Ortholog functional profiles of microbial communities was predicted by using PICRUSt (version 1.1.4) [36].

To statistically analyze the data, we used the online platform of Wekemo Bioincloud (https://www.bioincloud.tech/, accessed on 13 December 2023) [37]. The obtained data for the two groups (mycosphere soil and bulk soil) were consistent with the normal distribution, but the variance was unequal between groups. Their results were expressed as mean values, and these were compared between groups by Welch’s *t*-test [38]. 

Detrended correspondence analysis (DCA) results showed that the length of the first axis was below 3 [39]. So, based on the relative abundance of microbial species at different taxa levels, a redundancy analysis (RDA) was performed using the ‘vegan’ package in R to reveal the association between microbial communities and environmental factors [40]. Spearman’s correlation coefficients among the top 50 mycosphere’s bacterial genera and soil properties were calculated and displayed as a heat map by using R (version 4.2.3)’s ‘pheatmap’ package [36]. LEfSe was used to analyze the significance of differences in KEGG function between groups, and the results were visualized using bar charts [41]. The Spearman’s correlation analysis of soil properties and the diversity indexes were calculated by SPSS21.0.

## 3. Results

### 3.1. Site Sampling of Mycosphere and Bulk Soils

The soil organic carbon (SOC) content ranged from 69.16 to 176.23 g/kg across the 30 samples, whose soil pH value was between 5.71 and 6.47, while the soil water content (SWC) was 38.74–76.04%. The available nitrogen (AN) content was 174.07–320.6 mg/kg; available phosphorus (AP) content was 1.59–6.87 mg/kg; and available potassium (AK) content was 149.02–278.16 mg/kg. Hence, the soil physicochemical properties varied widely among 30 samples (Table 1).

The pH, SWC, SOC, and AN were significantly higher, but AP and AK were lower, in the mycosphere soil of *Calvatia candida* compared with bulk soil (*p* < 0.001) (Appendix A). In contrast, both AN and AK were significantly greater in the mycosphere soil than bulk soil of *Leucopaxillus laterarius* (*p* < 0.001), but vice versa for SWC, SOC, and AP (Appendix A). Compared with bulk soil, SWC, AN, AP, and AK were significantly increased in the mycosphere soil of *Leucopaxillus giganteus* (*p* < 0.001), but the pH and SOC were significantly decreased (Appendix A). In the mycosphere soil of *Lepista panaeola*, SWC, SOC, AN, and AP were all significantly higher than in its bulk soil, whereas pH and AK were significantly increased in the latter (*p* < 0.001) (Appendix A). 

To sum up, in the mycosphere soil of all samples, there was significantly greater AN content or SWC vis à vis bulk soil (*p* < 0.001). The AP content of mycosphere soil belonging to *Leucopaxillus giganteus* and *Lepista panaeola* was significantly higher than their bulk soil (*p* < 0.001). The AK was significantly higher in the mycosphere soil of *Leucopaxillus laterarius* and *Leucopaxillus giganteus* than their bulk soil (*p* < 0.001). The SOC was significantly higher in the mycosphere soil of *Calvatia candida* and *Lepista panaeola* than their bulk soil (*p* < 0.001). Accordingly, these results showed that compared with bulk soil, the mycosphere soil was richer in nutrients (Appendix A).

### 3.2. Bacteria Communities and Structure in the Mycosphere and Bulk Soils

After smoothing according to the minimum sample sequence, an average of 32,574 features sequences were obtained for each sample for further analysis (Figure 1). We used 15 mycosphere soil samples and 15 bulk soil samples of five ectomycorrhizal fungi to study the differences between their mycosphere and bulk soil bacterial communities. Compared with bulk soil, the Chao index for the mycosphere soil of *Calvatia candida*, *Russula brevipes*, *Leucopaxillus laterarius*, and *Leucopaxillus giganteus* was significantly lower (*p* ≤ 0.001), while that of *Lepista panaeola* was significantly higher (*p* ≤ 0.001) (Figure 2A). The Shannon index for the mycosphere soil of *Calvatia candida*, *Leucopaxillus laterarius*, and *Leucopaxillus giganteus* significantly exceeded that for their bulk soil (*p* ≤ 0.001), but vice versa for *Russula brevipes* and *Lepista panaeola* (*p* ≤ 0.001) (Figure 2B).

Among the five kinds of ectomycorrhizal fungi in mycosphere soil, *Calvatia candida* had the highest Chao index, followed by *Lepista panaeola*, *Russula brevipes*, and *Leucopaxillus giganteus*, being lowest in *Leucopaxillus laterarius* (Figure 3A). *Calvatia candida* also had the highest Shannon index, followed by *Leucopaxillus laterarius*, *Leucopaxillus giganteus*, and *Russula brevipes*, being lowest in *Lepista panaeola* (Figure 3B). Since both Chao and Shannon indexes of *Calvatia candida* were the greatest, this indicated that this fungus had the highest number of bacterial species and highest community diversity (Figure 3).

After smoothing, an average of 32,574 features sequences were obtained from the 30 samples of the five ectomycorrhizal fungi, which became clustered into 49 phyla. Proteobacteria, Acidobacteriota, Actinobacteriota, Bacteroidota and Verrucomicrobiota were the dominant phylum across the soil samples (Figure 4). 

Overall, 655 genera were found in the sequencing data, and the top 50 genera across all samples were analyzed further. *Solibacter* (*p* = 0.036), *Puia* (*p* = 0.016), and *Chthoniobacter* (*p* = 0.026) were significantly increased in the mycosphere soil of *Calvatia candida* vis à vis its bulk soil, whereas *Escherichia* (*p* = 0.035) was significantly decreased (Figure 5, Appendix A). Conversely, *Escherichia* (*p* < 0.001) was significantly increased in the mycosphere soil of *Russula brevipes* versus the bulk soil. Yet, in stark contrast, many more genera—*Usitatibacter* (*p* = 0.013), *Phenylobacterium* (*p* = 0.013), *Ferruginibacter* (*p* < 0.001), *Gaiella* (*p* = 0.012), *Hypericibacter* (*p* < 0.001), *Terrimonas* (*p* < 0.001), *Allosphingosinicella* (*p* = 0.035), *Solibacter* (*p* = 0.038), *Variovorax* (*p* = 0.016), *Ginsengibacter* (*p* = 0.047), *Devosia* (*p* = 0.017), *Gemmatimonas* (*p* = 0.011), *Methyloceanibacter* (*p* = 0.018), and *Niastella* (*p* = 0.015)—were increased significantly in the bulk soil of *Russula brevipes* relative to its mycosphere soil (Figure 5, Appendix A). *Escherichia* (*p* = 0.043) was significantly greater in the bulk soil of *Leucopaxillus laterarius* than its mycosphere soil (Figure 5, Appendix A). Four genera, *Pseudomonas* (*p* = 0.028), *Asprobacter* (*p* = 0.030), *Escherichia* (*p* = 0.014) and *Rudaea* (*p* = 0.049), were significantly increased in the mycosphere soil of *Leucopaxillus giganteus* compared with its bulk soil. *Escherichia* (*p* = 0.001) increased significantly in the bulk soil of *Leucopaxillus giganteus* versus mycosphere soil (Figure 5, Appendix A). *Escherichia* (*p* < 0.001) was significantly greater in the mycosphere soil than bulk soil of *Lepista panaeola* (Figure 5, Appendix A). *Sphingomicrobium* (*p* = 0.028), *Terrimonas* (*p* < 0.001), *Gemmatimonas* (*p* = 0.011), *Halomonas* (*p* = 0.027) and *Bryobacter* (*p* = 0.017) were significantly increased in the mycosphere soil of *Lepista panaeola* compared with its bulk soil (Figure 5, Appendix A).

Further comparison of the top 50 genera in the mycosphere soil of the five ectomycorrhizal fungi showed that those bacteria with a relative abundance of >1% in *Calvatia candida*’s mycosphere soil were *Usitatibacter* (3.45%), *Bradyrhizobium* (1.88%), and *Sphingomicrobium* (1.71%), of which *Usitatibacter* was dominant (Appendix A). Three genera also had a relative abundance of >1% in *Russula brevipes*’s mycosphere soil, namely *Escherichia* (30.62%), *Usitatibacter* (1.59%), and *Achromobacter* (1.62%), with *Escherichia* evidently dominant (Appendix A). In *Leucopaxillus laterarius*’s mycosphere soil, the genera with a relative abundance > 1% were *Escherichia* (1.07%), *Usitatibacter* (2.52%), *Bradyrhizobium* (1.70%), *Sphingomicrobium* (1.28%), *Phenylobacterium* (1.38%) and *Dongia* (1.25%), with *Usitatibacter* being dominant (Appendix A). Those genera with relative abundance >1% in *Leucopaxillus giganteus*’s mycosphere soil were *Escherichia* (2.43%), *Bradyrhizobium* (1.57%), *Pseudomonas* (4.93%), *Flavobacterium* (1.13%), *Pedobacter* (2.41%), *Brevundimonas* (2.02%), and *Halomonas* (1.59%), among which *Pseudomonas* dominated (Appendix A). In *Lepista panaeola*’s mycosphere soil, the genera with relative abundance > 1% were *Escherichia* (32.71%), *Achromobacter* (1.02%), *Bifidobacterium* (2.27%) and *Faecalibacterium* (1.35%), and *Megasphaera* (1.53%), with *Escherichia* as the dominant one (Appendix A). The genera with significant differences in mycosphere soil among the five ectomycorrhizal fungi were *Escherichia* (*p* < 0.001), *Usitatibacter* (*p* = 0.007), *Pseudomonas* (*p* = 0.002), *Pseudomonas* (*p* = 0.002), *Achromobacter* (*p* = 0.043), *Reyranella* (*p* = 0.004), *Solibacter* (*p* < 0.001), *Puia* (*p* = 0.016), *Rudaea* (*p* = 0.041), *Aquihabitans* (*p* < 0.001), and *Faecalibacterium* (*p* = 0.047) (Appendix A).

RDA can convey the relationships between bacterial taxa, samples, and the environment. Here, the RDA for soil bacteria from the five ectomycorrhizal fungi showed that the RDA1 and RDA2 axes, respectively, explained 38.24% and 22.36% of the overall community variance (Figure 6). The results showed that SWC (*r*^2^ = 0.452, *p* < 0.001), soil pH (*r*^2^ = 0.295, *p* = 0.012), and AK (*r*^2^ = 0.407, *p* = 0.001) were the main factors shaping bacterial communities (Figure 6).

The Spearman correlation analysis of the top 50 genus-level bacterial groups and environmental factors revealed that SOC and altitude clustered into one category, while AN and AK clustered into another category, with AP, SWC, and soil pH clustering into a single category (Figure 7). This indicated that AP, SWC, and pH had a relatively large impact on soil bacterial communities. Both *Pedobacter* (*p* = 0.029) and *Pseudomonas* (*p* = 0.035) had a significant positive correlation with AN (Figure 7). While *Bradyrhizobium* (*p* = 0.044) and *Ginsengibacter* (*p* = 0.009) each showed a significant negative correlation with AK, *Brevundimonas* (*p* = 0.028), *Aestuariivirga* (*p* = 0.018), *Bryobacter* (*p* = 0.010), *Chryseolinea* (*p* = 0.019), and *Flavobacterium* (*p* = 0.046) were all significantly positively correlated with AK (Figure 7). *Hypericibacter* (*p* = 0.023), *Devosia* (*p* = 0.014), and *Puia* (*p* = 0.014) had significant negative correlations with AP, while *Achromobacter* (*p* = 0.015), *Escherichia* (*p* = 0.018), and *Faecalibacterium* (*p* = 0.036) had significant positive correlations with AP (Figure 7). In relation to SWC, the genera *Gaiella* (*p* = 0.010), *Methyloceanibacter* (*p* = 0.038), *Mesorhizobium* (*p* = 0.001), *Reyranella* (*p* = 0.007), *Solibacter* (*p* = 0.013), and *Bradyrhizobium* (*p* = 0.021) were significantly negatively correlated with it, yet *Sphingobium* (*p* = 0.026), *Achromobacter* (*p* = 0.033), *Escherichia* (*p* = 0.002), *Bacillus* (*p* = 0.045), *Bifidobacterium* (*p* = 0.004), and *Megasphaera* (*p* = 0.005) were all significantly positively correlated with it (Figure 7). Finally, *Piscinibacter* (*p* = 0.016), *Dongia* (*p* = 0.015), *Rudaea* (*p* = 0.048), *Solibacter* (*p* = 0.037), *Usitatibacter* (*p* = 0.013), *Hypericibacter* (*p* = 0.049), and *Puia* (*p* = 0.003) were all significantly negatively correlated with pH, while *Pedobacter* (*p* = 0.001), *Brevundimonas* (*p* = 0.032), *Achromobacter* (*p* = 0.025), and *Escherichia* (*p* = 0.031) showed significant positive correlations with pH (Figure 7).

### 3.3. Functional Predictions in Mycosphere and Bulk Soils

The KEGG functions of the identified bacteria were checked for their significance (*p* < 0.05) in being affected mycosphere soil versus bulk soil (Figure 8). These results revealed that the ‘PI3K-Akt signaling pathway’ (*p* = 0.018), ‘Plant-pathogen interaction’ (*p* = 0.30), ‘Penicillin and cephalosporin biosynthesis’ (*p* = 0.031), ‘FoxO signaling pathway’ (*p* = 0.033), and ‘Mineral absorption’ (*p* = 0.044) were each significantly increased in the mycosphere soil of *Calvatia candida* (Figure 8). Conversely, ‘Fat digestion and absorption’ (*p* < 0.001), ‘Chronic myeloid leukemia’ (*p* = 0.028), ‘Base excision repair’ (*p* = 0.046), and ‘Notch signaling pathway’ (*p* = 0.048) were all significantly increased in the bulk soil of *Calvatia candida* (Figure 8). 

In the mycosphere soil of *Russula brevipes*, we found ‘Monobactam biosynthesis’ (*p* = 0.013), ‘ECM-receptor interaction’ (*p* = 0.018), ‘Dilated cardiomyopathy (DCM)’ (*p* = 0.027), ‘Systemic lupus erythematosus’ (*p* = 0.039), ‘FoxO signaling pathway’ (*p* = 0.040), ‘Taurine and hypotaurine metabolism’ (*p* = 0.042), ‘Arrhythmogenic right ventricular cardiomyopathy (ARVC)’ (*p* = 0.044), ‘Focal adhesion’ (*p* = 0.045), ‘Rap1 signaling pathway’ (*p* = 0.048), and ‘Selenocompound metabolism’ (*p* = 0.048) all significantly increased vis à vis bulk soil (Figure 8). 

For *Leucopaxillus laterarius*, in its bulk soil, both ‘Glycosaminoglycan degradation’ (*p* = 0.030) and ‘Fc epsilon RI signaling pathway’ (*p* = 0.049) were significantly increased relative to mycosphere soil (Figure 8). The ‘Sphingolipid signaling pathway’ (*p* < 0.001), ‘NOD-like receptor signaling pathway’ (*p* = 0.047), and ‘Antifolate resistance’ (*p* = 0.049) were increased significantly in the mycosphere soil of *Leucopaxillus giganteus* vis à vis bulk soil (Figure 8). ‘Glutamatergic synapse’ (*p* = 0.031), ‘Carbapenem biosynthesis’ (*p* = 0.036), and ‘Glyoxylate and dicarboxylate metabolism’ (*p* = 0.038) were all significantly increased in the mycosphere soil of *Lepista panaeola* (Figure 8). Conversely, in the bulk soil of *Lepista panaeola*, we found the following significantly increased: ‘Isoquinoline alkaloid biosynthesis’ (*p* = 0.015), ‘Phagosome’ (*p* = 0.033), ‘Various types of N-glycan biosynthesis’ (*p* = 0.036), ‘Gap junction’ (*p* = 0.038), ‘Glycosphingolipid biosynthesis-ganglio series’ (*p* = 0.038), and ‘Isoflavonoid biosynthesis’ (*p* = 0.041) (Figure 8).

The LEfSe analysis uncovered significant differences in the KEGG functions of 27 of the 5 ectomycorrhizal fungi in mycosphere soil (Figure 9). *Calvatia candida* mycosphere soil had three differential functions: ‘Biosynthesis of terpenoidsand steroids’ (*p* = 0.029), ‘Biosynthesis of unsaturated fatty acids’ (*p* = 0.046), and ‘Renin-angiotensin system’ (*p* = 0.047) (Figure 9). *Russula brevipes* mycosphere soil had three differential functions: ‘Signaling pathways regulating pluripotency of stem cells’ (*p* = 0.035), ‘alpha-Linolenic acid metabolism’ (*p* = 0.035), and ‘Riboflavin metabolism’ (*p* = 0.047) (Figure 9). In *Leucopaxillus laterarius* mycosphere soil, just one was differentiated, namely ‘Longevity regulating pathway-multiple species’ (*p* = 0.045) (Figure 9). *Leucopaxillus giganteus* mycosphere soil had seven differential functions, these being ‘D-Arginine and D-ornithine metabolism’ (*p* = 0.046); ‘Tyrosine metabolism’ (*p* = 0.047); ‘Biofilm formation-Pseudomonas aeruginosa’ (*p* = 0.021); ‘Fluorobenzoate degradation’ (*p* = 0.049); ‘Chlorocyclohexane, chlorobenzene degradation’ (*p* = 0.048); ‘Inositol phosphate metabolism’ (*p* = 0.048); and ‘Toluene degradation’ (*p* = 0.048) (Figure 9). The *Lepista panaeola* mycosphere soil was distinguished in having 13 differential functions: ‘Nitrotoluene degradation’ (*p* = 0.039), ‘Carbon fixation in photosynthetic organisms’ (*p* = 0.042), ‘Glycolysis and Gluconeogenesis’ (*p* = 0.037), ‘Pyrimidine metabolism’ (*p* = 0.049), ‘Pantothenate and CoA biosynthesis’ (*p* = 0.045), ‘Propanoate metabolism (*p* = 0.029), ‘C5-Branched dibasic acid metabolism’ (*p* = 0.039), ‘Type I diabetes mellitus’ (*p* = 0.031), ‘Purine metabolism’ (*p* = 0.048), ‘Nicotinate and nicotinamide metabolism’ (*p* = 0.045), ‘Methane metabolism’ (*p* = 0.034), ‘GABAergic synapse’ (*p* = 0.026), and ‘Insulin signaling pathway’ (*p* = 0.047) (Figure 9).

## 4. Discussion

Past studies have found that bacterial diversity in the mycosphere soil of most ectomycorrhizal fungi is lower than that in bulk soil [25,26]. For example, bacterial diversity was markedly lower in *Tricholoma matsutake*’s mycosphere soil than counterpart bulk soil (*p* < 0.05) [25]. Using R2A agar, the diversity of mycosphere soil bacteria of *Laccaria bicolor* was shown to be substantially lower than in corresponding bulk soil [42]. In the present study, the Chao index values in the bulk soil of *Calvatia candida*, *Russula brevipes*, *Leucopaxillus laterarius*, and *Leucopaxillus giganteus* were always significantly higher than in their mycosphere soil, and likewise for the Shannon index of the bulk soil of *Russula brevipes* and *Lepista panaeola* (Figure 2). This indicates that the mycosphere soil bacterial diversity of most ectomycorrhizal fungi studied here is lower than the bulk soil bacterial diversity, which is consistent with the results of previous studies [25,26]. In addition, the soil bacteria richness and community diversity in the mycosphere soil was slightly different among the five ectomycorrhizal fungi species (Figure 3). This indicated that ectomycorrhizal fungi differ in their ability to enrich and utilize bacteria [43]. The reduced bacterial diversity in the soil mycosphere may be caused by the dominance of fungal mycelium in its growth environment [44].

Proteobacteria, Acidobacteriota, Actinobacteriota, and Bacteroidota were the dominant phyla in the sampled soils (Figure 4). In the mycosphere and bulk soils of the five ectomycorrhizal fungi, the total relative abundance of those phyla (pooled) surpassed 84% (Figure 4). At the genus level, *Escherichia* (*p* < 0.05) is a significant group shared by the five ectomycorrhizal fungi in this study, and its content is significantly higher in the mycosphere soil of *Russula brevipes* and *Lepista panaeola* (Figure 4), which could increase the availability of soil phosphorus according to previous studies [45,46]. *Usitatibacter* not only has the ability to fix inorganic CO_2_ [47], but it also promotes the degradation of stubborn organophosphorus [48] and can be used as a biofertilizer to bolster crop yields [49]. *Bradyrhizobium* is a kind of nitrogen-fixing bacteria [50], which is capable of considerably increasing the nitrogen content of soil, along with contributing greatly to a higher soil phosphorus content [51]. This provides the necessary nitrogen and phosphorus for the growth of ectomycorrhizal fungi. *Pseudomonas* is an identified genus of mycorrhizal-helper bacteria (MHB) [52], and its functions are reflected in many aspects [12]. For instance, Shinde [53] found that the synergistic effect of *Pseudomonas* and *Laccaria bicolor* can promote the mycorrhization of *Populus* roots, and can augment the interaction between *Laccaria. bicolor* and *Populus*. In addition, a synergistic effect arising between *Pseudomonas* and *Laccaria trichodermophora* can increase the rate of the mycorrhizal colonization of *Pinus* tree roots to 93.5%–98.5%, and lead to *Pinus* tree obtaining a higher biomass [54]. *Pseudomonas* can also reduce the incidence of soil-borne diseases, increase crop yields, and promote the growth of host plants [55]. *Mesorhizobium* is a kind of rhizobia that functions as a nitrogen-fixer [56]. *Sphingomonas* also fixes nitrogen, in addition to two other functions: dissolving phosphate and producing plant growth hormones [57,58]. A study has demonstrated that *Gemmatimonas* members are deeply involved in phosphate and phosphite metabolism processes in soil [59]. *Acidoferrum* can enhance the Fe cycle and release inorganic P to support tree growth [60]. Chandni et al. [61] showed that *Devosia* can promote the degradation of organic compounds, aromatic compounds, and xenobiotic compounds in soil, as well as the decomposition of urea. *Chthoniobacter* is capable of degrading complex organic compounds [62], helping to improve soil fertility [63]. *Bacillus* can be used as a biofertilizer to inoculate functional bacteria so as to improve soil’s fertility and quality [64], with evidence that some strains can promote the growth and development of large fungi [43]. *Sphingobium* has abundant xylan-degrading enzymes that can degrade biological macromolecules, cellulose, hemicellulose, and xylan, all of which are beneficial for the growth and development of large fungi [65]. In concert, these bacteria play an important role in the growth of ectomycorrhizal fungi; thus, we speculate that they are the MHB of ectomycorrhizal fungi. In this study, we also noticed that the MHB in the mycosphere soil of various ectomycorrhizal fungal species were quite different (Appendix A). Different ectomycorrhizal fungi may prefer or rely on certain specific MHB [13], such that some ectomycorrhizal fungi may form closer symbiotic relationships with particular strains or species of MHB.

As the growth environments of mycelium, ectomycorrhizal and mycosphere are the biological and abiotic factors affecting the soil ecosystem [66,67,68]. Ectomycorrhizal fungi mostly grow in forest environments with a large plant coverage, a rich humus layer, and fertile soil [69]. The application of biochar and mycorrhiza not only increases the content of soil organic carbon [70], total nitrogen, total phosphorus and other nutrients [71], but also increases the availability of soil phosphorus [72,73,74]. A higher nitrogen content can stimulate ectomycorrhizal fungi to produce more spores, making it easier for them to colonize symbiotic plants [75]. In our study, the AN content of the five ectomycorrhizal fungi’s mycosphere soil significantly exceeds that of their bulk soil, with the AP content of the mycosphere soil belonging to *Leucopaxillus giganteus*, and *Lepista panaeola* being significantly higher than that of their bulk soil, and likewise for AK with respect to *Leucopaxillus laterarius* and *Leucopaxillus giganteus* (Table 1). Compared with bulk soil, SOC is significantly higher in the mycosphere soil *Calvatia candida* as well as *Lepista panaeola* (Table 1). Altogether, these results show that the growth of ectomycorrhizal fungi is capable of augmenting soil nutrient contents, which is consistent with previous reports [76,77,78,79]. This suggests that the symbiosis between ectomycorrhizal fungi and plants could improve the nutrient status of mycosphere soil. 

Our RDA results showed that three abiotic factors, namely pH, SWC, and AK, have significant effects on the soil bacteria of ectomycorrhizal fungal species (Figure 6). Some work has shown that ectomycorrhizal fungi like acidic soil, and the most suitable pH value for their growth is 4.0–6.0 [80]; the soil pH value in our study is 5.71–6.47, which differs little from that, supporting this conclusion. Furthermore, the soil pH value can strongly affect soil bacterial community diversity [81,82], with the influence of ectomycorrhizal fungi on soil bacteria regulated, in part, by soil pH [83]. Research has shown that the mycosphere soil bacterial community of ectomycorrhizal fungi is significantly affected by SWC [79]. Bacteria not only passively respond to soil moisture but can also actively change soil water-related properties, such as water permeability, water-holding capacity, and evaporation [84]. Some soil characteristics (such as AN and SOC) are directly or indirectly related to soil pH, which may lead to corresponding changes in the bacterial community structure [85]. The effects of altitude on soil bacterial communities in the mycosphere of five ectomycorrhizal fungi were not significant, likely due to the small difference in elevation across the sampling area.

PICRUSt2 was used to perform the KEGG functional annotations for the ectomycorrhizal fungi’s mycosphere soil bacteria. These results demonstrate that mycosphere soil functions are different from bulk soil functions. The growth of soil bacteria is mainly achieved by ingesting nutrients such as amino acids, carbohydrates, and vitamins in the soil [86]. The root secretions of ectomycorrhizal fungi contain many organic nutrients [86], which provide vital nutrients for soil bacteria. Accordingly, the carbon source [87] regulates the relationship between ectomycorrhizal fungi and soil bacteria [25], thus contributing to the enrichment of soil bacterial communities [88]. In our study, we also detected divergent soil functions of the mycosphere among ectomycorrhizal fungal species. We hypothesize that this might be due to the biased selection of soil bacteria by different ectomycorrhizal fungi, which can lead to differing bacterial community functions. These differences may in turn influence various characteristics of the five species of ectomycorrhizal fungi and their growth and development in the wild.

Ectomycorrhizal fungi, as an important part of the ecosystem [89], together with mycosphere soil bacteria, constitute a complex microbial network [19], which plays a crucial role in maintaining the function and stability of the ecosystem [2]. Through the comparative analysis of five kinds of ectomycorrhizal mycosphere soil, we found that there were significant differences in soil bacterial diversity in the mycosphere of different ectomycorrhizal fungi, which directly or indirectly led to the formation of different soil functions in mycosphere soil of ectomycorrhizal fungi. Through a systematic analysis of environmental factors, we also found that soil conditions (SWC, pH, and AK) have a significant impact on the composition of soil bacterial communities in mycosphere, emphasizing that environmental factors should be considered in the soil management of ectomycorrhizal fungi.

## 5. Conclusions

By comparing the physicochemical properties as well as bacterial diversity and community structure of the five ectomycorrhizal fungi’s mycosphere soil and bulk soil, the environment suitable for the growth of each fungus could be determined. The 16S rRNA sequencing results show that the bacterial community composition of mycosphere soil differs significantly from that of bulk soil. Further analysis shows that the growth of ectomycorrhizal fungi results in changes in microbial community structure. The growth of ectomycorrhizal fungi reduces the diversity and abundance of soil bacterial communities. Among the soil variables, SWC, pH and AK contributed significantly to the bacterial community structure and diversity characteristics in all the geographical regions where ectomycorrhizal fungi were studied. *Escherichia*, *Usitatibacter*, *Bradyrhizobium*, *Pseudomonas*, *Mesorhizobium*, *Sphingomonas*, *Gemmatimonas*, *Acidoferrum*, *Devosia*, *Chthoniobacter*, *Bacillus*, and *Sphingobium* are potential mycorrhizal-helper bacteria (MHB) of ectomycorrhizal fungi. These different ectomycorrhizal fungi may selectively promote the growth and reproduction in soil of specific MHB in their symbiosis system, forming a mutualistic ecosystem. In addition, the presence of different ectomycorrhizal fungi may lead to the activation or inhibition of specific metabolic pathways in soil, thus affecting the diversity and functional potential of soil bacterial communities. Therefore, via an in-depth understanding of the interactions between ectomycorrhizal fungi and mycosphere soil bacteria, more effective soil management measures could be designed for practical mushroom production, including the use of tailored biofertilizers containing MHB, and so on, to promote the protection, artificial cultivation, and sustainable use of ectomycorrhizal fungi.

## Figures and Tables

**Figure 1 microorganisms-12-01329-f001:**
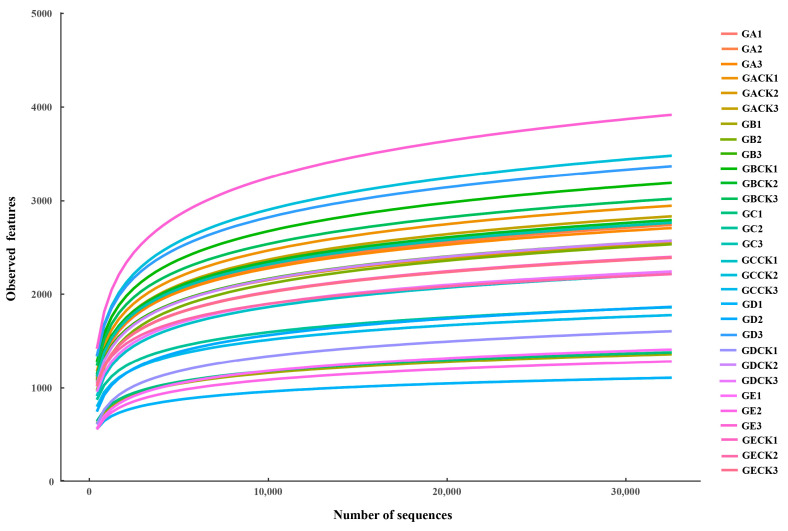
Dilution curves for bacterial signature sequences. GA: *Calvatia candida*’s mycosphere soil, GACK: *Calvatia candida*’s bulk soil, GB: *Russula brevipes*’s mycosphere soil, GBCK: *Russula brevipes*’s bulk soil, GC: *Leucopaxillus laterarius*’s mycosphere soil, GCCK: *Leucopaxillus laterarius*’s bulk soil, GD: *Leucopaxillus giganteus*’s mycosphere soil, GDCK: *Leucopaxillus giganteus*’s bulk soil, GE: *Lepista panaeola*’s mycosphere soil, and GECK: *Lepista panaeola*’s bulk soil.

**Figure 2 microorganisms-12-01329-f002:**
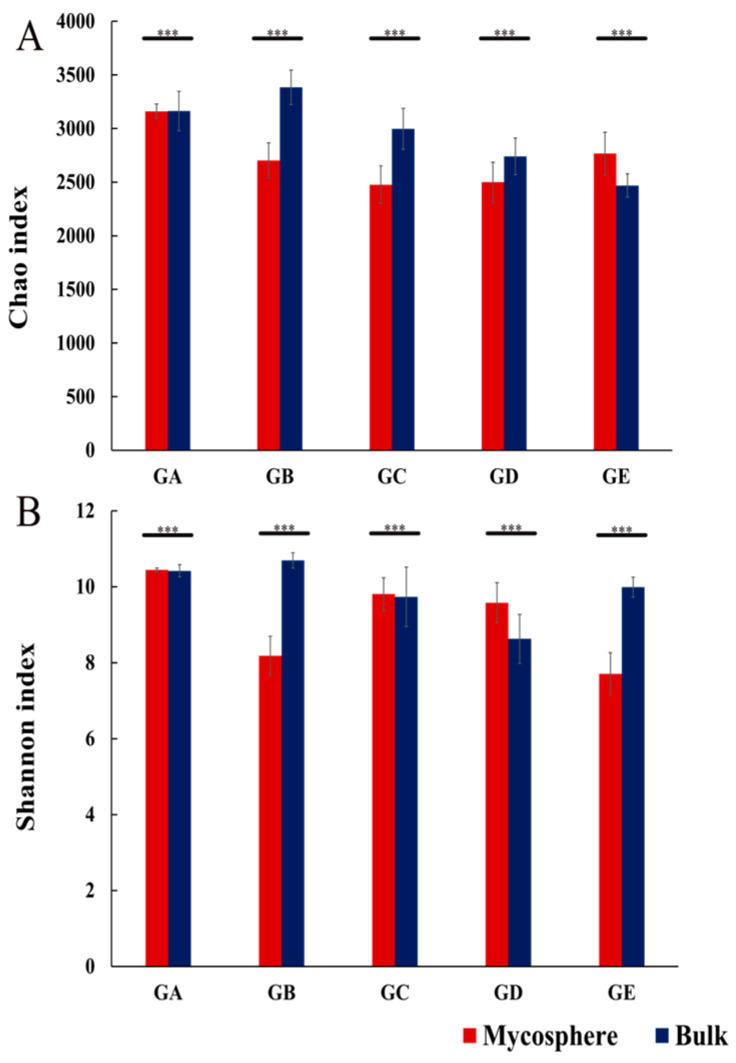
Comparison of Chao (**A**) and Shannon (**B**) indexes between mycosphere and bulk soil. GA: *Calvatia candida*, GB: *Russula brevipes*, GC: *Leucopaxillus laterarius*, GD: *Leucopaxillus giganteus*, and GE: *Lepista panaeola*. Significant differences by *** *p* ≤ 0.001. Data are mean ± SE (*n* = 3).

**Figure 3 microorganisms-12-01329-f003:**
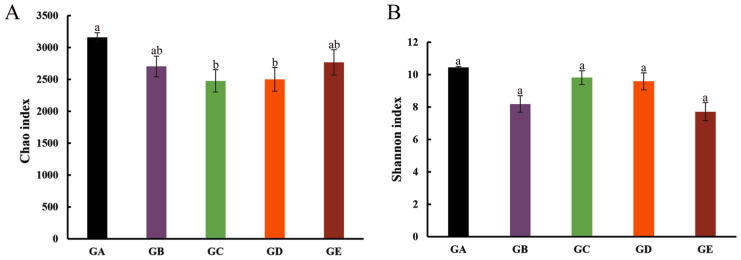
Comparison of Chao (**A**) and Shannon (**B**) indexes in mycosphere soil of five ectomycorrhizal fungi. GA: *Calvatia candida*, GB: *Russula brevipes*, GC: *Leucopaxillus laterarius*, GD: *Leucopaxillus giganteus*, and GE: *Lepista panaeola*. Data are mean ± SE (*n* = 3). Different lowercase letters indicate significant differences (*p* < 0.05).

**Figure 4 microorganisms-12-01329-f004:**
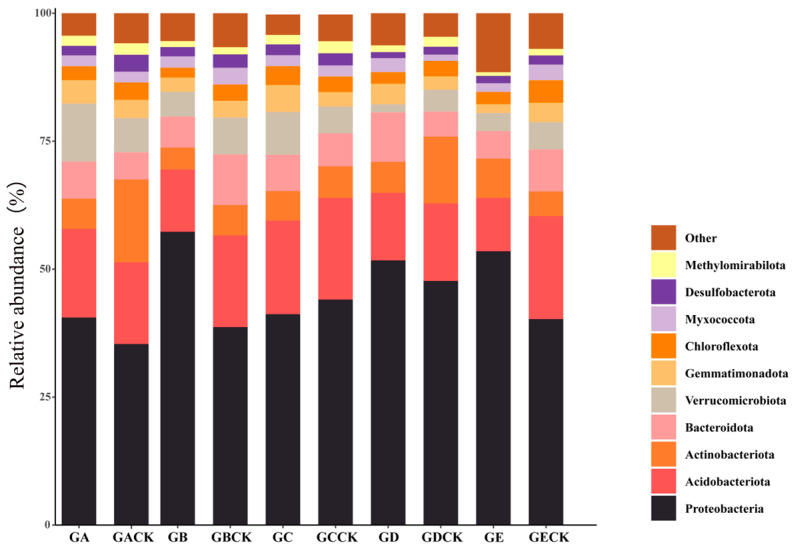
Comparison of phyla between mycosphere and bulk soil. GA: *Calvatia candida*’s mycosphere soil, GACK: *Calvatia candida*’s bulk soil, GB: *Russula brevipes*’s mycosphere soil, GBCK: *Russula brevipes*’s bulk soil, GC: *Leucopaxillus laterarius*’s mycosphere soil, GCCK: *Leucopaxillus laterarius*’s bulk soil, GD: *Leucopaxillus giganteus*’s mycosphere soil, GDCK: *Leucopaxillus giganteus*’s bulk soil, GE: *Lepista panaeola*’s mycosphere soil, and GECK: *Lepista panaeola*’s bulk soil.

**Figure 5 microorganisms-12-01329-f005:**
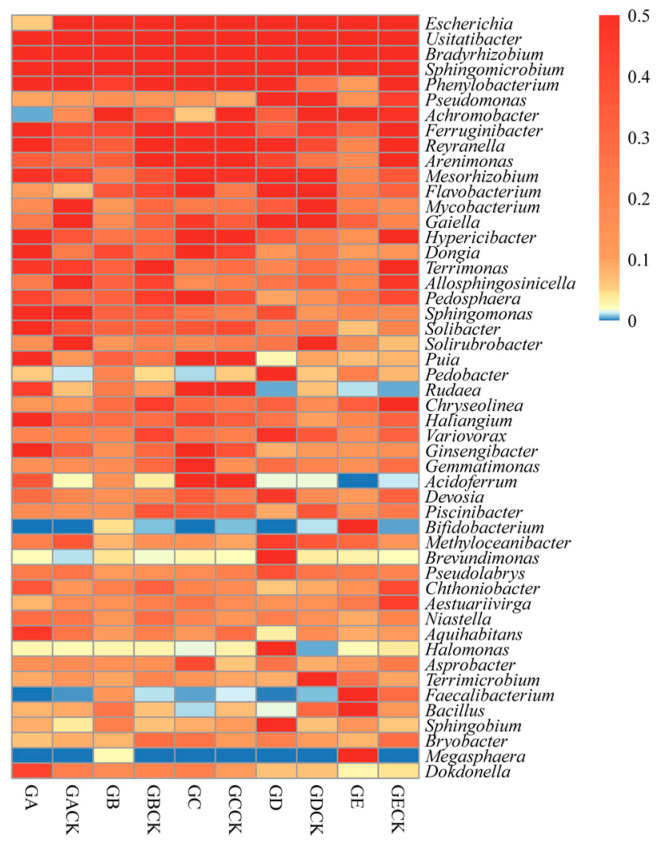
Community composition of five species of ectomycorrhizal fungi mycosphere and bulk soil bacteria at the top 50 genera level. GA: *Calvatia candida*’s mycosphere soil, GACK: *Calvatia candida*’s bulk soil, GB: *Russula brevipes*’s mycosphere soil, GBCK: *Russula brevipes*’s bulk soil, GC: *Leucopaxillus laterarius*’s mycosphere soil, GCCK: *Leucopaxillus laterarius*’s bulk soil, GD: *Leucopaxillus giganteus*’s mycosphere soil, GDCK: *Leucopaxillus giganteus*’s bulk soil, GE: *Lepista panaeola*’s mycosphere soil, and GECK: *Lepista panaeola*’s bulk soil.

**Figure 6 microorganisms-12-01329-f006:**
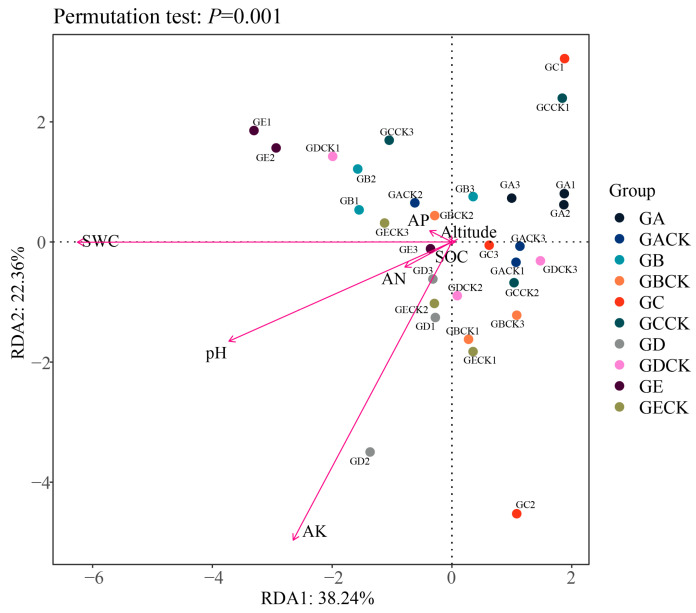
Redundancy analysis of the top 50 genera and environmental factors. GA: *Calvatia candida*’s mycosphere soil, GACK: *Calvatia candida*’s bulk soil, GB: *Russula brevipes*’s mycosphere soil, GBCK: *Russula brevipes*’s bulk soil, GC: *Leucopaxillus laterarius*’s mycosphere soil, GCCK: *Leucopaxillus laterarius*’s bulk soil, GD: *Leucopaxillus giganteus*’s mycosphere soil, GDCK: *Leucopaxillus giganteus*’s bulk soil, GE: *Lepista panaeola*’s mycosphere soil, and GECK: *Lepista panaeola*’s bulk soil.

**Figure 7 microorganisms-12-01329-f007:**
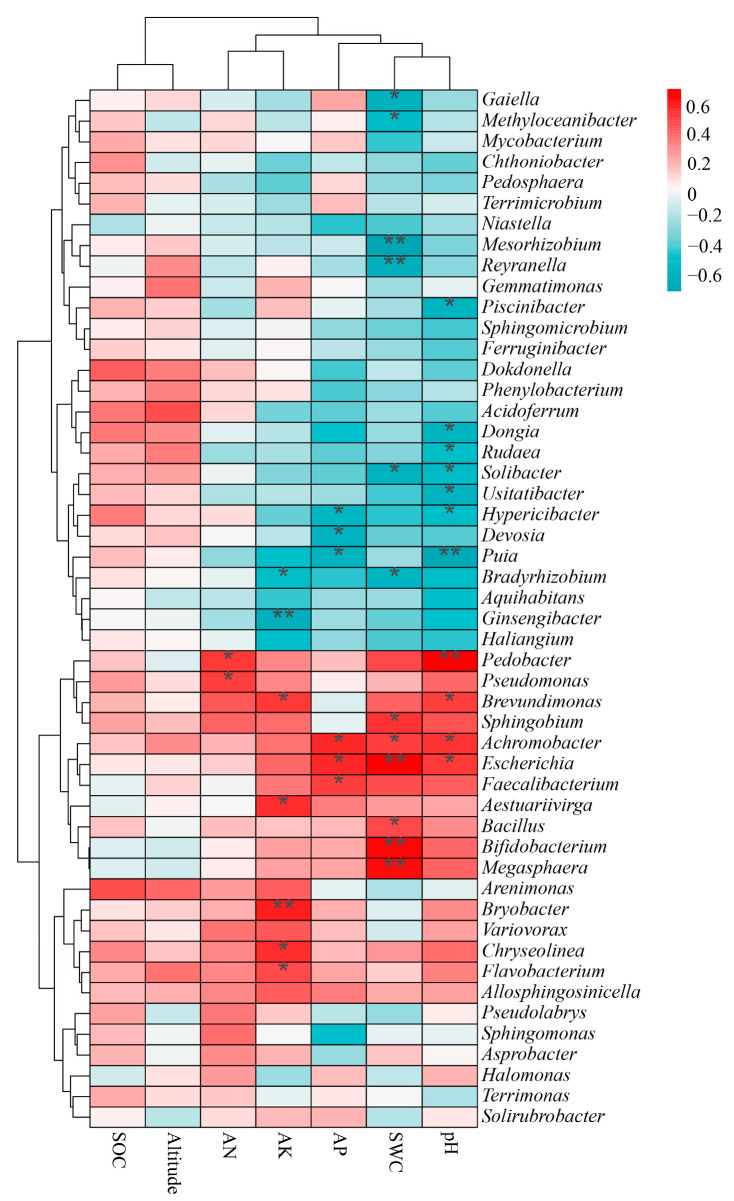
Spearman correlation between the top 50 genera and environmental factors. Significant differences by * 0.01 < *p* ≤ 0.05. ** 0.001 < *p* ≤ 0.01.

**Figure 8 microorganisms-12-01329-f008:**
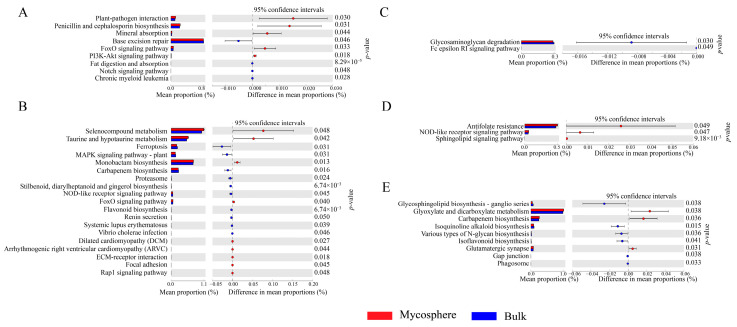
Comparison of the KEGG function between mycosphere and bulk soil. (**A**) *Calvatia candida*, (**B**) *Russula brevipes*, (**C**) *Leucopaxillus laterarius*, (**D**) *Leucopaxillus giganteus*, and (**E**) *Lepista panaeola*.

**Figure 9 microorganisms-12-01329-f009:**
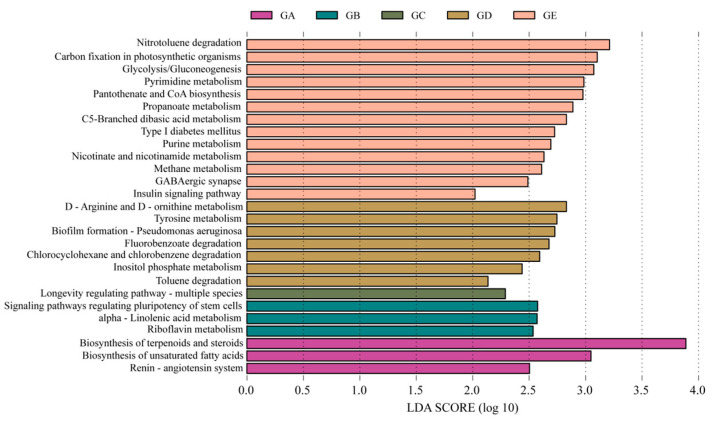
The five kinds of ectomycorrhizal fungi have significantly different functions in mycosphere soil. GA: *Calvatia candida*, GB: *Russula brevipes*, GC: *Leucopaxillus laterarius*, GD: *Leucopaxillus giganteus*, and GE: *Lepista panaeola*.

**Table 1 microorganisms-12-01329-t001:** Site information used for this study.

Sample	ECM Species	Replicate	Longitude (E)	Latitude (N)	Altitude (m)	pH	SWC (%)	SOC (g/kg)	AN (mg/kg)	AP (mg/kg)	AK (mg/kg)
GA	*Calvatia candida*	3	111°27′21″	37°49′24″	1886.54	5.94	43.63	121.36	257.13	1.91	165.31
GACK		3				5.87	38.74	110.72	234.27	2.18	234.53
GB	*Russula brevipes*	3	111°27′8″	37°49′28″	1922.88	6.33	76.04	240.73	320.60	5.03	263.75
GBCK		3				6.03	60.47	176.23	254.33	4.14	278.16
GC	*Leucopaxillus laterarius*	3	111°27′9″	37°51′45″	1908.7	5.71	39.69	84.18	180.13	5.81	273.31
GCCK		3				6.12	46.65	99.75	174.07	6.87	149.02
GD	*Leucopaxillus giganteus*	3	111°27′7″	37°51′46″	1896.51	6.44	50.29	80.66	297.50	6.31	273.07
GDCK		3				6.49	46.49	96.62	193.67	1.59	180.93
GE	*Lepista panaeola*	3	111°26′24″	37°53′8″	1886.33	6.21	73.95	71.16	212.80	3.71	241.80
GECK		3				6.47	64.12	69.16	179.67	3.66	270.35

SOC, SWC, AN, AP, and AK represent soil organic carbon, soil moisture content, available nitrogen, available phosphorus, and available potassium, respectively. All values are averages. GA: *Calvatia candida*’s mycosphere soil, GACK: *Calvatia candida*’s bulk soil, GB: *Russula brevipes*’s mycosphere soil, GBCK: *Russula brevipes*’s bulk soil, GC: *Leucopaxillus laterarius*’s mycosphere soil, GCCK: *Leucopaxillus laterarius*’s bulk soil, GD: *Leucopaxillus giganteus*’s mycosphere soil, GDCK: *Leucopaxillus giganteus*’s bulk soil, GE: *Lepista panaeola*’s mycosphere soil, and GECK: *Lepista panaeola*’s bulk soil.

## Data Availability

The raw reads were deposited into the NCBI Sequence Read Archive (SRA) database (accession number: PRJNA1121956).

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
