# Peer review of "Comparison of Bacterial Communities in Five Ectomycorrhizal Fungi Mycosphere Soil"

_microorganisms, 2024, doi:10.3390/microorganisms12071329_

Round 1
Reviewer 1 Report
Comments and Suggestions for Authors
The manuscript with the title “Comparison of bacterial communities in five ectomycorrhizal fungi mycosphere soil” provides an analysis of mycosphere soil of five endomycorrhizal fungi by offering new insights into bacteria community structure and physical-chemical parameters of the soil that might be in relationship with functional roles of these fungi.
The introduction is adequate for the topic. However, besides mentioning bacteria at lines 49-50, it does not clarify for the reader what is the current knowledge on the topic, nor if anything is known on the potential relationship between bacteria and fungi. Based on the literature research authors must be able to explicitly say something such as “Although ectomycorrhizal fungi can host bacteria, few studies addressed …” or alternatively “little is known about … therefore, this research addresses a gap of knowledge regarding …”. Such an approach is very important for the justification of the study, and immediately highlights is relevance and novelty.
Species names with italics. E.g. Line 138, 144, 149, 151, 179-184 … species are with regular font.
Table 1 shall contain a footnote explaining the acronyms because table should be self-explanatory.
Figure 2 – what are the whiskers representing?
Figure 3 – what are the whiskers representing? Please mention how significance was assigned in the caption – the p threshold or test (still p ≤ 0.001 or p 0.05?)
Lines 198- 217, Lines 221-242, Lines 251-272 Bacteria genera with italics.
Figure 6 – caption replace “50 genus” with “50 genera”
Figure 8 components are difficult to read.
Discussion is well written. However, it is not clearly explained how this research can improve our current knowledge and what brings new. Only the last part of the conclusions section addresses this aspect and advise authors to also insert something in the discussion section. I suggest authors to insist on these new highlights and matters.
Best regards.
Comments on the Quality of English Language
moderate English improvements - grammar and syntax are recommended.
Author Response
Commtnts 1: The introduction is adequate for the topic. However, besides mentioning bacteria at lines 49-50, it does not clarify for the reader what is the current knowledge on the topic, nor if anything is known on the potential relationship between bacteria and fungi. Based on the literature research authors must be able to explicitly say something such as “Although ectomycorrhizal fungi can host bacteria, few studies addressed …” or alternatively “little is known about … therefore, this research addresses a gap of knowledge regarding …”. Such an approach is very important for the justification of the study, and immediately highlights is relevance and novelty.
Response 1: Done. We added this section in lines 50-62 and highlighted it in red.
Commtnts 2: Species names with italics. E.g. Line 138, 144, 149, 151, 179-184 … species are with regular font.
Response 2: It has been amended and the full text has been checked.
Commtnts 3: Table 1 shall contain a footnote explaining the acronyms because table should be self-explanatory.
Response 3: We've added footnotes explaining these acronyms, highlighted in red on lines 171-173. We've also added explanations for all figure headings so that you just understand.
Commtnts 4: Figure 2 – what are the whiskers representing?
Response 4: We added this to the caption of the image on line 197.
Commtnts 5: Figure 3 – what are the whiskers representing? Please mention how significance was assigned in the caption – the p threshold or test (still p ≤ 0.001 or p 0.05?)
Response 5: This section was added in lines 208-209.
Commtnts 6: Lines 198- 217, Lines 221-242, Lines 251-272 Bacteria genera with italics.
Response 6: Done.
Commtnts 7: Figure 6 – caption replace “50 genus” with “50 genera”
Response 7: Done.
Commtnts 8: Figure 8 components are difficult to read.
Response 8: Done. We adjusted Figure-8.
Commtnts 9: Discussion is well written. However, it is not clearly explained how this research can improve our current knowledge and what brings new. Only the last part of the conclusions section addresses this aspect and advise authors to also insert something in the discussion section. I suggest authors to insist on these new highlights and matters.
Response 9:Done. We have added some content on lines 465-474 and highlighted it in red.

Reviewer 2 Report
Comments and Suggestions for Authors
Manuscript entitled <Comparison of bacterial communities in five ectomycorrhizal fungi mycosphere soil>
present useful information on ectomycorrhizal fungi in China.It aims at solve Past findings that bacterial diversity in the mycosphere soil of most ectomycorrhizal fungi is lower than that in bulk soil. For example, in some Tricholoma mycosphere.
the manuscript needs to be improved:
table1:
Add: ECM species
fig 4: to better highlight control (bulk soil)
use scientific names in italics, along the manuscript
In short, the manuscript presents usefull information; however, it is a little hard to read.
Please, name controls, explain more sample collection. Did you use a transec?
Fig.4: to better highligth control/bulk soil
Author Response
Commtnts 1: table1: Add: ECM species
Response 1: Done. We added footnotes to explain ECM species.
Commtnts 2: fig 4: to better highlight control (bulk soil)
Response 2: Done, we have added an explanation in the title of figure-4.
Commtnts 3: use scientific names in italics, along the manuscript
Response 3: Done.
Commtnts 4: Please, name controls, explain more sample collection.
Response 4: Done. We have added explanations to the titles of all the figures for your better understanding.
Commtnts 5: Fig.4: to better highligth control/bulk soil
Response 5: Done.

Round 2
Reviewer 2 Report
Comments and Suggestions for Authors
The manuscript was improved accordingly to suggestions.
Author Response
Done.